# Barriers and Mythical Practices of Teenagers Regarding the Prevention of Sexually Transmitted Infections in Rural Areas of Limpopo Province, South Africa

**DOI:** 10.3390/healthcare12030355

**Published:** 2024-01-30

**Authors:** Jessica Uchechi Damian, Eustacia Hlungwane, Takalani Grace Tshitangano

**Affiliations:** 1The Best Health Solutions, Johannesburg 2198, South Africa; damjessicauche@gmail.com; 2Department of Public Health, Faculty of Health Science, University of Venda, Thohoyandou 0950, South Africa; eustaciaeusy@gmail.com

**Keywords:** barriers, Limpopo, measures, mythical, practices, prevention, rural, STIs, teenagers, villages

## Abstract

Sexually transmitted infections in South Africa are increasing at alarming rates. This study’s setting is no. 5, with the highest STI, pregnancy, and HIV statistics in Limpopo province among 13- to 19-year-old teenagers. This study explored preventative measures against STIs practiced by teenagers in rural areas of Limpopo province, South Africa. This study was conducted at a selected rural-based clinic using an exploratory descriptive qualitative research approach. Unstructured in-depth face-to-face interviews were used to collect data from sixteen conveniently sampled teenagers aged 13–19, consisting of 13 females and 3 males, who came to Manavhela Clinic for youth-friendly services in August/September 2022. Open-coding analysis was used to identify themes and sub-themes. Measures to ensure trustworthiness were ensured. Ethical clearance (FSH/21/PH/22/2211) was obtained, and ethics principles were observed throughout this study. Two themes emerged from data analysis: STI preventive measures practiced by teenagers and factors influencing the choice of STI preventive measures practiced by teenagers. Only a few participants aged 13 and 14 years of age practiced abstinence and condom use. Most participants were sexually active and used mythical mixtures made from boiling aloe or morula tree (which they drank before and after sex), applied plain yogurt on the vagina once a week, or practiced vaginal steaming. Participants cited patriarchy, lack of sex education in rural schools, long distances to clinics, and desire to taste sex as reasons for adopting the practiced preventive measures. Risky sexual behavior among 13- to 19-year-old teenagers is still rife in rural areas. Rural clinics in Limpopo province should intensify STI school health education and youth-friendly services programs to raise awareness and improve accessibility to condoms.

## 1. Introduction

Worrying trends in the sexual well-being of adolescents are globally increasing the prevalence rates of teenage pregnancy and STIs in specific regions [1]. Sexually transmitted infections (STIs) remain a public health burden to both individuals and the health care sector globally [2]. According to the World Health Organization (WHO) [2] fact sheet, there are more than a million estimated daily diagnoses of asymptomatic STIs globally. If untreated, they can have a detrimental effect on the sexual and general health status of individuals of all ages. An estimated 96 million incident cases of syphilis, gonorrhea, chlamydia, and trichomoniasis have been recorded among people between the ages of 15–49 years in the African region. STIs have been identified as a high-risk determinant for human immunodeficiency virus (HIV) transmission, adult T-cell leukemia, cancers, infertility, and adverse pregnancy outcomes such as pre-term birth, stillbirth, and congenital deformities [2,3].

In the South African context, studies [4,5,6] demonstrated the high prevalence of STIs in both rural and urban settlements. Limpopo province, located in north-eastern South Africa with a population of 5.9 million people with the majority mainly from the Pedi, Tsonga, and Venda tribes [6], has also experienced an increased prevalence rate among patients attending with STI symptoms in some districts. Thus, in 2017/2018, the Capricorn District had an STI rate of 69.4% compared with Mopani (68.6%), Sekhukhune (68.5%), and Waterberg (98%) [1]. Adolescents and teenagers have been recognized as one of the priority populations that are at risk of becoming infected with STIs due to possible risky sexual behaviors [3,7,8], which may be because of some personal (for instance, sex and age of individual (it is perceived that females tend to engage in sexual activities earlier than males)) or social (for instance, social media networks (it is perceived that social media networks makes it easier for teens to share sexual contents among themselves and also search for supposedly “reliable” information about sex)) influences as implied by Larsson, Bowers-Sword, Narvaez, Ugarte [9], and Nunu, Makhado, Mabunda, and Lebese [10] Based on this evidence, actions have strategically taken place in different countries based on WHO guidelines to combat the incidence and prevalence of STIs, beginning with the priority groups [1,3].

As an intervention, the South African government launched the National Adolescents and Youth-Friendly Service (NAYFS), which extended to the provincial Departments of Health, including Limpopo [11,12]. The NAYFS program aimed to promote the health and well-being of youth aged between 10 and 24, especially at primary health care (PHC) facilities [11,12]. To achieve the purpose of the program, trained peer educators situated at the designated health facilities give information on topics such as “Know your body, rights, and responsibilities of adolescents and youths, benefits of abstinence, information on HIV and AIDS, medical male circumcision, and contraceptives” [11]. This National Adolescent and Youth-Friendly Service (NAYFS), which is guided by the Adolescent and Youth Health Policy of South Africa, aims to promote the health and well-being of young people aged 10–24 years by providing comprehensive integrated sexual and reproductive services, encouraging health seeking behaviors among youth, and empowering health providers to be advocates for youth. NAYFS is expected to meet the following ten standards in each facility in the province and the country:Management system support for the effective provision of adolescent- and youth-friendly health services.Policies and processes that support the rights of adolescents.Appropriate adolescent health services are available and accessible.The clinic has a physical environment conducive to the provision of adolescent-friendly health services.The clinic has adequate drugs, supplies, and equipment necessary to provide the essential service package for youth-friendly health care.Provision of relevant information, education, and communication (IEC) promoting behavior change consistent with the YFS essential service package.Systems in place to train and develop staff to provide effective adolescent-friendly health services.Adolescents receive adequate psycho-social and physical assessments.Adolescents receive individualized care based on standard case management guidelines/protocols.The clinic provides continuity of care for adolescents, i.e., proper referral systems are in place.

The key settings for the implementation of NAYFS include home, school, health facilities, workplace, community-based organizations, residential centers, and streets. Peer educators are the first point of contact for students seeking advice on health and other social issues at any setting. Manavhela Clinic is among the health facilities rendering such services.

Problem statement

Despite the availability of the National Adolescents and Youth-Friendly Service (NAYFS), there is still an increase in the number of new STIs recorded in Limpopo province, with rural clinics having the highest statistics [1,12]. In addition, there are socio-economic and media-circulating myths that limit the recognition of STIs as an essential public health problem and the uptake of evidence-based preventive measures. Furthermore, condom use and the associated behavioral change are low in rural areas [1], which is cause for great concern, because complications of STIs can cause health problems like cervical cancer, pelvic pains, and infertility [6]. The second author in this study is a professional nurse who works at the participating clinic in Limpopo province and noticed that more teenagers came to the clinic for the treatment of STIs than for condoms, which triggered a quest to conduct a study aimed at exploring the preventative measures practiced by teenagers against STIs.

Purpose

This paper investigates preventive measures that teenagers practice against STIs in rural Limpopo province, South Africa. In order to achieve the aim, this study explores the preventive measures practiced by teenagers against STIs, and also describes the factors that influenced their choice of STI preventive practices.

Definition of concepts

STIs, in this study, refer to HPV, HIV, syphilis, gonorrhea, chlamydia, trichomoniasis, and other infections acquired after having unprotected sex. STI preventive practices: in the context of this study, this refers to abstinence, post-exposure prophylaxis, condom use, and other non-specified measures. Teenagers: in the context of this study, this can also be used interchangeably with adolescents, referring to young people between the ages of 13 and 19.

## 2. Materials and Methods

Consolidated criteria for reporting qualitative research (COREQ) [13] was applied to describe the methods used in this study.

Study design

This study employed an exploratory and descriptive qualitative research design to help better understand and describe the preventative measures practiced by teenagers against STIs [13].

Study setting

This study was conducted at Manavhela Clinic. Manavhela Clinic is located within the Collins Chabane Municipality in South Africa, with a total population of 347,975 people residing in the municipal area [7]. The clinic falls under the Bungeni Local Area. Manavhela Clinic serves six communities: Manavhela, Tshitungulwane, Tshilaphala, Tshivhulana, Makhadzi, and Hasani-Dakari. It is situated next to Nzhwelule Primary School and a football field. The clinic is located 52 km away from Makhado and 28 km from Tshilidzini, which is the referral hospital. The clinic renders 24 h services through an on-call system from 18 h 00 to 07 h 00 a.m. daily. The services covered include tuberculosis (TB), HIV and AIDS, treatment of minor ailments, antenatal care (ANC), and maternity deliveries, as well as youth-friendly services (YFS) (that deal with family planning and reproductive education among teenagers). Most clients served by Manavhela Clinic depend on social grants for survival; some of these clients who visit the clinic belong to the Christian faith, while some belong to the traditional faith.

Population, sampling technique, and sample size

The target population for this study was teenagers aged 13–19 years who came to Manavhela Clinic for youth-friendly services, irrespective of gender. The average number of teenagers who visited Manavhela Clinic between March and July 2020 for different health care services was 105.

About 97 potential participants were approached face-to-face for recruitment to participate in this study. Thus, the participants’ selection used non-probability accidental/convenience sampling [14]. All those who were approached were given information about the researcher, the purpose of this study, their rights and expected responsibilities, the safety and security of their responses, the use of recording devices, and the possibility of anonymously publishing findings. Participants were made to understand that their participation in this study was voluntary, and they could opt out anytime, even after giving consent. About 81 teenagers refused to participate in this study, not citing any specific reasons. Only sixteen teenagers voluntarily agreed to participate. Thus, the sample size was only 16 participants: 2 aged 13 years, 2 aged 14 years, 2 aged 15 years, 3 aged 16 years, 2 aged 17 years, 3 aged 18 years, and 2 aged 19 years.

Inclusion and exclusion criteria

This study included 12 consenting teenagers who came to the youth-friendly service at Manavhela Clinic during August and September 2022. Teenagers who visited the clinic but were not part of the youth-friendly service at Manavhela Clinic were intended to be excluded. However, when data saturation was not attained based on the study set objectives, the researcher ultimately included four teenagers (p3, p7, p 10, and p16) who only attended the youth-friendly service one after the other until data saturation was achieved.

Data collection tool

Based on the study objectives, the researcher developed an interview guide in the English language and had it translated into Tshivenda and Xitsonga by language experts, after which it was vetted by the research supervisors and pre-tested to ensure that the data collection tool conveyed its intended meaning and achieved the research objectives. The data collection tool contains two central questions, namely: (a) What measures do you take to prevent STIs? (b) Why do you use those measures? These questions were followed by open-ended probing questions guided by the participant’s response. For the participants who were uncomfortable with the English language, the interviewer provided the questions in their preferred mother tongue (Tshivenda or Xitsonga). Participants responded in Tshivenda.

Pre-testing

Authors pre-tested the research questions to check whether they were clear enough to provide detailed information and whether they were not leading to a “yes” or “no” answer. The voice recorder was connected to check its functionality and that of the batteries to ensure that the process of collecting data was not interrupted. The researcher allocated three (more than 10%) participants from the target population who were first met and interviewed to check the effectiveness of the interview guide. In so doing, the researcher could check the interview time, resolve the ambiguity of the questions, and make appropriate adjustments. All participants in pre-testing also formed part of the final study to ensure that the essential data provided during the pre-testing phase were retained.

Measures to ensure trustworthiness.

Credibility, transferability, dependability, and confirmability were adopted to ensure this study’s trustworthiness. This study’s credibility was ascertained through prolonged engagement with the study participants, data collection through audio-recorded interviews and field notes for data triangulation, and member-checking by replaying recorded interviews for each participant after the interview process to verify the accuracy of the researcher’s interpretation. A detailed description of the research methodology process was documented and followed strictly to ensure the transferability of this study’s findings. For the dependability and confirmability of this study to be ensured, all transcripts and the voice recorder were made available to the supervisor (third author) and co-coder (first author) to confirm the findings. The data collected went through an audit trail by two expert researchers in the form of a supervisor and co-supervisor to check and compare the results.

Ethical consideration

Before participant recruitment, an ethical clearance certificate was obtained from the University of Venda Human and Clinical Research Ethics Committee (FSH/21/PH/22/2211), while permission to access the selected clinic and participants was issued by the Limpopo Department of Health (LP-2021-12-001) and the operational manager of Manavhela Clinic.

To avoid bias, as the researcher is a professional nurse in the participating clinic, the recruitment process for potential participants was conducted by trained home-based carers and non-governmental organization personnel who came to the clinic every week. Participants were recruited while facilitating youth-friendly services at Manavhela Clinic. Participants were informed of the interview’s allocated time frames before the interviews. Written consent forms were obtained from sixteen teenagers who agreed to participate in this study.

Data were collected during consultation time; no one at the clinic knew the specific day for data collection. Voices were lowered during data collection so that people from other cubicles could not hear anything. Due to the sensitive nature of this study, the questions were tailored to ensure that the researcher minimized sensitive issues, with the help of the supervisors, who are experts in the field. It is noteworthy to point out that during the interview process, the researcher professionally asked questions. Therefore, no harm occurred during data collection, and the researcher did not experience emotional harm feedback from the participants. Codes were assigned to participants’ responses in the field notes instead of their names to maintain confidentiality and anonymity.

Data collection process

The female researcher (masters of public health student at the University of Venda), the second author, collected data at the clinic in a separate consultation room, where there were no other people but one participant at a time and the researcher. Rapport was created prior to asking serious study questions. The process of data collection took two months (August and September 2023) after approval by the University Research Ethics Committee. To ensure that data of good quality were collected during the interview process, field notes were taken, data were summarized to condense and clear the participants’ statements, probing questions were asked to request more information during the interview, and active listening was practiced, as suggested by Grey et al. [15]. The interviews per participant lasted between 15- and 30-min. Data were saturated at Participant 14, but the researcher continued until Participant 16 to confirm the saturation. There were no repeat interviews carried out.

Data analysis

The second author conducted thematic open coding analysis [15] to transform raw data into themes and sub-themes according to meaning and relevancy to the study objectives. These codes were confirmed by the co-coder, who is the third author. Before the analysis, some of the interviews recorded in Tshivenda were transcribed verbatim into English by a language expert. Afterward, all the transcripts were aligned with the field notes and were returned to participants, but there were no comments or corrections. Codes, themes, and sub-themes were generated using thematic analysis.

## 3. Results

Table 1 below depicts the demographic characteristics of the sixteen teenagers who participated in this study.

Open coding thematic analysis yielded two themes tabulated in Table 2 below.

### 3.1. STI Preventative Measures Practiced by Teenagers

Abstinence

Analysis of the data revealed that only 2 out of the 16 participants mentioned that they abstain from sex as an STI preventive measure.

Participant 1, a 13-year-old female, said “*I just abstain from sexual activities to prevent myself from contracting sexually transmitted infections.*” Another one explained

“*I’m still young, so abstaining from sexual activities is the best way for me to avoid being affected by the STIs. However, with our generation, it is difficult to meet these requirements. It is just difficult to abstain, I am saying this because many teenage girls get pregnant these days which will come as a shock to a lot of people*”.(Participant 3, 13-year-old female)

Condom use

Some participants mentioned that they engaged in sexual activities but that they did not consistently use condoms for protection. Participant 5 (female, 15 years old) said:

“*I use protection some other times and sometimes I do not. On most occasions, I use protection; on some, I can say I did not. Some of these things you cannot control it. Sometimes you are caught in a situation where you end up doing something you never planned to do*”.

Another participant also said:

“*I usually use a condom when I plan to have sex, but in case of emergency, I don’t because I didn’t have time to prepare for sex it just happened*”.(Participant 11, female, 15 years old)

Mythical practices

Participants explained some of the mythical practices they adopt before, during, and after sex as a form of preventive measures against sexually transmitted infections.

“*I boil aloe (tree) and drink the water before and after sex, and I will also give it to my girlfriend to drink so that we will be protected and safe from the STIs*”.(Participant 9, male, 14 years old)

“*I boil the morula tree (stem) before sex and drink the water while it is warm. This will prepare my body to resist the possibilities of being affected and that my partner is also protected*”.(Participant 7, male, 14 years old)

“*If it is an emergency, my boyfriend will release (ejaculate) outside my vagina to protect and keep us safe from the STIs*”.(Participant 11, female, 15 years old)

Participant 4 and 16 (two females, 18 years old) said “*I use nothing; after having unprotected sexual intercourse with someone, I will then use plain yogurt to wash and cleanse my vagina, or sometimes steam my vagina using salty warm water and put it in the bucket, and sit on top of it, all the discharges will be wiped away*”.

### 3.2. Factors Influencing the Choice of STI Preventive Practices

Lack of sex education

Some participants pointed out that having knowledge about sexually transmitted infections and practicing it would protect teenagers from contracting sexually transmitted infections and would also help curb the incidence of teenage pregnancy.

Participant 16 (female, 18 years old) said “*I think some of us, we end up using the traditional methods we are told of preventing the spread of STIs because we do not know about the prevention of STIs, let alone to know what the STIs are. I use concoctions because of the first-hand information I get at home and from my schoolmates*”.

Participant 6 (female, 14 years old) said “*I follow safer practices to prevent the contraction of STIs when engaging in sexual activities because I do know how one can be contracted with the STIs and how it could be spread from one person to another. To my understanding, I think most of my friends could take education and schooling very seriously some of the things that happen will not happen, such as teenage pregnancy*”.

Participant 2 (female, 17 years old) suggested that having a better orientation and intervention strategy for STI preventive measures may help to embolden teenagers, especially those in the formal education setting, to act right without fear of judgment. Her words:

“*I think there is a need to rethink a strategy on how to encourage teenagers to use the appropriate measures to prevent them from contracting and spreading STIs. The availability of condoms at school does not mean that all the teenagers who are going to school will use them, some are shy to take them while they are being watched by others which results in them using the whatever methods that we are not sure if they are to assist*”.

Peer pressure

Some participants explained that they experience much pressure from their peers and have access to wrong information.

“*I use concoctions because is the first-hand information that I get at home and from my schoolmates*”.(Participant 16, female, 18 years old)

“*I sometimes don’t use protection I can also say that I get influenced by my friends on the measures we can use to prevent ourselves from contracting the STIs*”.(Participant 11, female, 17 years old)

“*There are some other stories, [based on a myth that we might be holding], that we might be sharing for fun as teenagers when we are alone, and these stories are powerful in that we will want to try these things when we are alone since we want to experience stuff*”.(Participant 11, female, 15 years old)

Accessibility of sexual health services

Some participants reported that, even though they were aware of the STI preventive measures, they lived very far away from hospitals or primary health care facilities where they could easily have access to condoms. Participants 5 (male, 15 years old) and 10 (male, 18 years old) had this to say:

“*…… we end up using the traditional methods we are told of prevents the spread of STIs because we do not have access to the condoms*”.

“*……. but also, because we are far from the clinics where we can get the information and other services such as the PrEP pills you can get after you think that you might have been exposed to HIV/ AIDS. More health workers should come to our community to educate more people regarding the transmission of STIs and how we can protect each other*”.

Gender Power Dynamics

This study revealed that the issue of gender power dynamics is one of the factors that determines the preventive measure to be adopted by some teenagers. Only one participant pointed it out and had this to say:

“*I don’t usually use any other means of protection. If my boyfriend says we should use a condom we do that, if he says we should not use it we simply don’t, don’t forget that males are the head of the family we as females should support whatever they say. My boyfriend has a final say on our sex life*”.(Participant 13, female, 19 years old)

## 4. Discussion

This study was set to investigate the preventative measures practiced by teenagers against STIs and the factors influencing the choice of STI preventive practices in rural areas. Findings were organized into:

### 4.1. STI Preventative Measures Practiced by Teenagers

The following preventive measures were found to be practiced by teenagers:

Abstinence

The findings of this study revealed that few, especially the youngest of the participants (aged 13 years), practiced abstinence as their preventive measure against STIs. These findings are supported by Irfan, Hussain, Noor et al. [16], who discovered a prevalence of 3.4–83.3% primary abstinence among young men (10 to 24 years old) globally, with Nigeria having a prevalence of abstinence among male and female youth (16–24 years old) of 68%. Similar findings in Kwazulu Natal, South Africa, by Zuma, Seeley, Sibiya et al. [17] indicated that abstinence and being faithful were uncommon prevention strategies and were used mainly by young girls who went to study away (sent by parents to a boarding school) and those who practiced virginity testing because they delayed starting sexual relationships. The perception was that these two approaches are not easy to adhere to, as most young people in the community start having relationships and experimenting with sex when they reach secondary school. The findings of this study suggest that even though 14 out of the 16 participants were part of NAYFS at the participating clinic (where the benefits of abstinence have been taught), there is inadequate translation of knowledge to practice regarding the use of sexual abstinence as a preventive measure against STIs. In addition, whilst in this study sample abstinence was rarely encountered, it is important to note that the sample was from teenagers attending the clinic and was not a community sample. The findings simply suggest that even teenagers at a young age have sexual contact. A community sample is needed to determine the extent of this behavior in the community.

Condom use

The findings of this study indicate that those few participants who used condoms used them inconsistently. Ajayi, Omonaiye and Nwogwugwu et al. [18] found supporting evidence that indicated that the prevalence of consistent condom use was 39.3% (CI: 35.5–43.2%) among the Eastern Cape Province university students, with no significant gender and age differences. Similarly, Duby, Jonas, McClinton Appollis et al. [19] highlighted that the use of condoms amongst youth in South Africa is still suboptimal. A similar finding was highlighted by Mostert, Sethole, Khumisi, Peu, Thambura et al. [20], who revealed low (41.2%) use of condoms among youth in Northwest province, South Africa. Low and inconsistent condom use are both risky sexual behaviors that might be contributing to STIs and teenage pregnancy in South Africa.

Mythical preventive practices

Findings in this study revealed that some participants used mythical mixtures such as a drink made from boiling aloe or marula tree (which they drank before and after sex), applying plain yogurt on the vagina once a week, or vaginal steaming. Furthermore, studies have considered the application of yogurts as beneficial against women’s vaginal diseases. Johannsen [21] asserted as early as 2005 that placebo-controlled cross-over trials performed by Hilton and collaborators using L. acidophilus-containing yogurt had a cure rate ranging from 57% for bacterial vaginosis to 74% for yeast vaginitis compared with a cure rate of 0–22% observed in the control group. In addition, Johannsen [21] stated that Israeli researchers using yogurt containing L. acidophilus found a decrease in the episodes of bacterial vaginosis among the participants ingesting it. These studies were also mentioned by Das and Ameeruddin [22], who confirmed that the application of yogurt is a scientific preventive measure and not a myth. However, since there is no current robust scientific evidence to support this practice, it is still mythical and cannot be recommended for health education prior to robust scientific evidence.

Regarding drinking boiled aloe juice before and after sex, Taylor-Donald [23] cited a study conducted by Russians that claims that aloe vera stimulates the normal protective function of the body and increases the body’s ability to handle harmful substances. According to Taylor-Donald [23], those who do make a daily habit of drinking aloe juice say that they have increased energy, their digestion is improved, and they feel generally healthier. If these people believe that aloe will help the body rid itself of ulcers, constipation, colitis, and arthritis (to name just a few), drinking aloe juice may be one of the best preventive measures. In addition, Olariu [24] claims that aloe moves into the body where it works to boost the body’s repair system, feeding every single cell with as many of the nutrients as it can, and that it is this action that leads to claims of a “curative” effect on arthritis, Candida, and nerves. Aloe vera is a wonder plant that influences any part of the human body through its healing power since it is a natural fighter against all sorts of infection [24]. However, Olariu [24] emphasizes that it might take weeks for these effects to be achieved, making the safety of sporadic drinking of aloe juice before and after sex a questionable myth that may not be promoted in facilities prior to future robust research.

Regarding the drinking of boiled marula tree leaves before and after sex, Foods and Abdalbasit [25] assert that a drink made from marula leaves is used to treat gonorrhea in Sudan. In contrast, the bark is also a disinfection when soaked in boiling water. According to Kyazike [26], syphilis can be cured by drinking the boiled roots of the marula tree mixed with those of ntwa and mulaliki. Thus, currently, using boiled leaves of marula trees might be a mythical preventive practice that might be contributing to increased STIs and pregnancies in Limpopo province.

### 4.2. Factors Influencing the Choice of STI Preventive Measures Practiced by Teenagers

Gender Power Dynamics

This study found that patriarchal relationships, which may be perceived as a cultural norm, put the participants in a situation where they indulge in unprotected sex because they are obliged to do as their male partner decides. Torregosa, M. and Patricio [27] highlight that ascribed gender roles, such as in the African culture, may influence relationship dynamics, decisions, and negotiations and, therefore, the practice of protected safe sex. According to Torregosa, M. and Patricio [27], African cultural norms assume women to be naive, submissive, nurturing, selfless, self-sacrificing, family-centered, chaste, and sexually pure, and female virginity is emphasized. The submissive nature of African women perpetuates the inhibition of expression and poor direct communication about sex protection between partners, which puts women at high sexual risks. Similar findings in Mozambique by Gruenbaum, Earp, and Shweder [28] discovered that girls were less assertive, more accepting of gender power differentials, and tended to be dependent on partners for material needs, which served to weaken their bargaining power concerning safe sexual behavior, which rendered them more vulnerable. This study’s findings have implications for the development of culturally sensitive evidence-based interventions that promote sexual self-efficacy skills among males who strongly identify with culturally ascribed gender roles.

Lack of sex education

This study also found that the lack of sex education is another factor influencing teenagers to indulge in unsafe sex. Leung, Shek, and Leung’s [1] review on existing sexuality education showed that many gaps and inadequacies in sex education exist (for example, lack of evidence-based policies and lack of multi-disciplinary collaboration in designing sex education programs) in two of the largest English-speaking countries (such as United States of America) and three Chinese-speaking societies (Mainland China, Hong Kong, and Taiwan). A similar finding was reported by Kyilleh et al. [29], where participants showed a lack of comprehensive knowledge of sexual reproductive health, which made them vulnerable to unsafe sexual reproductive health behavior and choices, with unplanned pregnancies and STIs as the outcomes of their choices. These findings imply that schools need to establish and strengthen reproductive health clubs to be able to equip students with the required skills and knowledge about sexuality.

Peer pressure

The findings of this study indicated that some of our participants’ choices of STI preventive measures were greatly influenced by their schoolmates and home environment, emphasizing the need for NAYFS to equip households and schools with sexuality education. Similarly, Girmay and Mariye [30] found that only 68 (60.8%) initiated their first sex of their own will in Ethiopian youths aged between 15 and 24 years. According to Adegboyega, Ayoola, and Muhammed [31], adolescents are more responsive to the rewards of risk, such as peer approval, because they are still developing their capacities for judgment and self-control. Peers were influenced by written messages, such as positive and negative reviews, comments, suggestions, discussions, or experiences [31]. A similar study by Moore and Rosenthal [32] found that 61 percent of sexually active young people received a large amount of their sexual education from discussions with friends only. Wakasa, Oljira, and Demena [33] suggested that this might be because adolescents are at a higher likelihood of sharing their day-to-day life experiences with their friends and that adolescents need attention and recognition with peers so that they are likely to behave in a manner that intimate friends practice. These findings imply that parents should also be part of NAYFS to make them aware of the dynamic behavioral change of their children and encourage them to listen to them and correctly attend to their sexual education needs.

Accessibility of sexual health services

This study revealed that the inability to access health facilities at crucial times (for example, when they need to have access to condoms) tends to be a hindrance to consistent compliance with safe sex practices, especially for those who do not attend NAYFS. This is because even services like having access to free packs of condom are usually made available in school settings through NAYFS, and those who live far from the health facilities can easily have access to pills like post-exposure prophylaxis that can prevent HIV infection if they were exposed. Ninsiima et al. [34] concurred with the findings of this study, indicating that the access and utilization of youth-friendly sexual and reproductive health remains a big problem for youth, especially in sub-Saharan Africa. Thus, about 20 studies from 7 sub-Saharan countries identified structural barriers such as negative attitudes of health workers and lack of knowledge among youth regarding YFSRHS as barriers to accessing sexual health services [34]. This study has implications for relevant stakeholders to implement quality implementation guidelines in clinics to offer services according to youths’ needs and preferences. Educating the youth in schools can facilitate the utilization and scale-up of NAYFS.

## 5. Conclusions

Despite their participation in NAYFS, risky sexual behaviors among 13 to 19-year-old teenagers reported by participants suggest that these behaviors could still be rife in rural areas, rooted in so-called rational ideas from scientific evidence, gender power dynamics, and possible inadequate use of NAYFS. Rural clinics in Limpopo province should intensify STI school health education and youth-friendly services programs to raise awareness and improve accessibility to condoms. The development of evidence-based sexuality education guidelines and messages for primary health care personnel, including community health workers, who respond to mythical-related information, would go a long way to empower youth in rural communities. Life orientation teachers strengthen the assertion and resilience skills of learners at school to resist peer pressure. Community health workers and NAYFS peer educators should educate more people in the community regarding the transmission and prevention of STIs, including sex negotiation and assertion. The Department of Cooperative Governance and Traditional Affairs should develop interest groups to consider policy development and guidelines that advocate for a shift from the dominant stereotypes about “backward” African cultures of gender power dynamics. Researchers should develop culturally sensitive evidence-based interventions that promote sexual self-efficacy skills among males who strongly identify with culturally ascribed gender roles. There is also a need to conduct this study with a large sample size in order to determine the prevalence level of these risky behaviors among teenagers in Vhembe District.

## Figures and Tables

**Table 1 healthcare-12-00355-t001:** Participant demographic characteristics.

Age Range	Gender	Total
13–15	Female	5
	Male	1
16–19	Female	8
	Male	2

**Table 2 healthcare-12-00355-t002:** Themes and sub-themes.

Themes	Sub-Themes
**STI preventative measures practiced by teenagers**	Abstinence Condom use Rational ideas based on evidence
**Factors influencing the choice of STI preventive measures practiced by teenagers**	Lack of sex education Peer pressure Gender power dynamics Accessibility of sexual health services

## Data Availability

Data for this study are available upon request from eustaciaeusy@gmail.com.

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
