# Peer review of "Barriers and Mythical Practices of Teenagers Regarding the Prevention of Sexually Transmitted Infections in Rural Areas of Limpopo Province, South Africa"

_healthcare, 2024, doi:10.3390/healthcare12030355_

Round 1
Reviewer 1 Report
Comments and Suggestions for Authors
The manuscript "Barriers and mythical practices of teenagers regarding sexually transmitted infections prevention in rural areas of Limpopo Province, South Africa" has merit and should be considered for publication, pending revisions. There remain few studies exploring in the barriers and practices of rural youth, and the impact of efforts to address their unique health burdens. This paper contributes to that literature. Importantly, the paper is also written by a scholar who is intimately familiar with the population and the context. This presents as an insider perspective. While this is one of the paper's strengths, there are moments when context needs more explanation.
Overall, the approach of the study is sound. Having only a few participants is a limitation, but it is more informative to speak to a few people who are interested in talking about the details of a shared practice, rather than having a larger number of people who provide only superficial answers. The challenge is ensuring that you don't assume that these 16 represent the entire group. So statements like "most" need to be deleted. What is uncovered is the scope of themes that resonate in driving or preventing safer sex practices, but not the extent to how many youth in this region practice those behaviors.
The authors mention the National Adolescents and Youth Friendly Service that is practicing in Limpopo clinics, and indicate that they had wanted to recruit only participants engaging with this program. Provide more detail about this program. What is the role of the youth in this program? What role do peer educators play? Are they only active inside the clinic or do they hold events or provide information in schools or in churches as well? Bring this program back into the discussion to evaluate its efficacy against the findings of the study. If there remains myths and behaviors that are not supported by sound scientific evidence, then is this program working?
Some minor edits:
Line 42: remove the word "young" to describe people age 15-49, as it is confusing with youth or teenagers that are your study population
Line 52: what are the "personal or social influences" that are "implied" in the literature
Line 69: what are the myths that circulate in the media?? and later on, how does the media influence the behaviors of youth, most mentioned getting information from friends
Write your objectives and definitions as part of the text rather than as bullets.
Materials and Methods section -- there is a lot of repetition here. Eliminate anything that appears twice, including the intentions of the researchers, the pre-testing section, and the data collection process.
The inclusion/exclusion section if very confusing. It almost reads like some of the participants were not engaged with the National Adolescents and Youth Friendly Service.
Line 111: Sentences needs to be reconstructed. It implies that people who are Christians do not receive social services
Line 113: The setting describes the clinic. This sentence about the habits of young people is out of place and not relevant to this study.
Table 1. Teenagers who visit the clinic, should correspond in time to the dates of the study. This way we can determine if the 97 teenagers that were approached were representative of the clinic population. Providing data that is 1 year different is not helpful. Also, the table isn't necessary. A monthly average of clinic patient number is adequate.
The translation of the questions into English could be clearer: The translation could be clearer: What measures or strategies do you take to prevent STIs? Why do you use those strategies?
Also, did the participants respond in English or were their answers translated? Is there any potential for mistranslation or confusion or multiple meanings?
Results:
Table 2 should be presented as a traditional demographics table that summarizes the participants not presents each individually (gender -- count of males and females; Age -- mean and range).
Table 3 is unnecessary
You don't need to provide details of the participants twice, either introduce them or include gender and age following the quote. Provide more than one person to support each idea. As it is, for each theme only one person is presented. This suggests that no one else mentioned that point. If others don't have compelling quotes, then indicate specific words they used to describe the idea or what specific elements they mention. For example, did only one female talk about applying yogurt and steaming? While we are not interested in counts, we do need to have some appreciation for how much these ideas were shared by the collective.
Line 251: "indulge" should be replaced by "practice" it is less judgmental. I also wonder if "mythical" is the correct term to use. In the discussion you present evidence to indicate that much of the "mythical" ideas are supported by data. Whether it is sound and robust scientific data is unclear, but it certainly suggests that these are not "myths" but rational ideas based on evidence.
Line 273: could you provide more details on where the young people in this study obtained their information. I realize it was not a primary study question, but it is frequently mentioned. What would happen if the information from home or friends conflicted with the formal information provided by the peer educators of the national program?
Discussion:
Line 336: virgin testing -- could you provide more information, is this still commonly practiced in the region? was it mentioned by anyone in the study?
Line 257: "most participants" as you have not recovered quantitative data, you need to take care with word choice here. You also only provide 1-2 quotes from individuals.
Line 389: revise the wording here. It is clear that there is not currently robust scientific evidence to support his practice. Could you also provide some indication of how valid the data if that you present to support the other practices. It seems you often present only one reference. Is that sufficient to conclude that the practice is supported by evidence?
Line 395: Patriarchy may not be the correct word. Patriarchy refers to male inheritance lines, male dominance in government. Gender power dynamics is a better way to describe the situation as it is presented.
Line 415: Can you provide details of this study, what are the gaps, what are the countries? What makes this study particularly relevant?
Line 421: you say that schools need to be better equipped, but you don't specifically include schools in the study. Did students say they didn't get information from school? How does the national program described in the introduction fit into education practices? Does it only provide information to youth who engage with the health clinics?
Line 437: again I think you overstep here in implicating parents. Unless you present data that specific indicates that parents are not a source of information or that youth want more information from their parents, drawing this conclusion from your study is not appropriate.
Accessibility of sexual health services -- relate this to the national program that all the participants were involved in. Were they not receiving adequate services through this program? were condoms not available? were they not comfortable talking to the peers or other educators? how does this program fit into this conclusion?
Comments on the Quality of English Language
Generally, the English was sound throughout the document. However, the study questions could be revised for clarity in addition to one or two important sentences (inclusion/exclusion criteria) and the methods section made more concise.
Author Response
Dear Reviewer
All comments made were appreciated and corrected.

Reviewer 2 Report
Comments and Suggestions for Authors
This study contributes to the literature and understanding of the sexual behaviour and practices of adolescents in the African context. However, as a qualitative study, with a very small sample, it is important that the results are not used to estimate the prevalence of the practices or behaviours in the community. Some of the interpretation of the results is based on implied prevalence of behaviours – while it would be more appropriate to focus on what the study reveals of the understanding of the subjects.
More detailed comments and suggestions for the authors follows.
Abstract
Does not mention the gender of the subjects
.Subjects stated as those aged 13 to 19 years – but gender distribution not mentioned.
“Risky sexual behaviour .. is still rife”. Important to clarify that this qualitative study cannot estimate the prevalence of the ideas or practices – but rather explores the thinking behind the behaviours. Perhaps a more appropriate summary would be that belief in traditional or “natural” remedies is still strong, rather than scientific or western medical concepts.
Introduction
Line 48: Limpopo has also experienced . For readers not familiar with South Africa, it would be helpful to provide some further information on Limpopo – its geography, population etc
Line 49 : These are presumably districts within Limpopo – do they cover all the districts of Limpopo ?
Line 49 – 50 :It is unclear what the percentage figures refer to - % increase (compared to previous year ?)
Line 74: Is the clinic in Limpopo province ?
Materials and methods
Line 99-101: This comment repeats the comments in line 73-74 of the introduction – it is probably unnecessary to repeat it here, while retaining the comments in lines 73-74
Line 103: It would be useful to include an estimate of the population in the clinic catchment, and/or the proportion or number of teenagers in the population.
Line 110: ‘productive education’ – probably should be ‘reproductive’
Line 140-143: It is unclear what this sentence means. The first sentence states that teenagers who were not part of the Youth Friendly Service were excluded; and the second that 4 teenagers who ‘only’ attended the Youth Friendly Service were included. Did they only attend the Youth Friendly Service after interviews ?
Line 152: ‘trial-ran’ – suggest using the term ‘pre-tested’
Line 189: ‘who experts’ - add ‘are’
Results
Table 3 provides a useful summary of the key results
4. Discussion
Line 327 Abstinence: while in this study sample, abstinence was rarely encountered, it is important to note that the sample was from teenagers attending the clinic and is not a community sample. It suggests that even teenagers at a young age have sexual contact, but qualitative studies cannot provide prevalence estimates, and a community sample is needed to determine the extent of this behaviour in the community.
Similarly, line 339 also attempts to draw prevalence estimates from this qualitative study, which is not appropriate.
The discussion of the effectiveness of yoghurt application to the vagina (lines 359-365) the consumption of ale vera juice (lines 368 – 381) or drinking of marula ( lines 383 ff) extends beyond the scope of the study on practices and behaviours. It would be more relevant to compare these beliefs and practices with the recommendations and advice that should be provided by the health or education facilities.
4.2 Factors influencing choice of STI preventive measures
Lack of sex education (lines 414 ff). While the stated beliefs of the participants suggest a lack of sex education, there is no reporting of questions about sex education or access to sex education materials in the results section. It is perhaps an assumption.
Accessibility of sexual health services. This is perhaps an issue that requires further research, as the study participants were accessed through a sexual health service, and thus do not represent the views of those who do not access such services.
5. Conclusions
‘Risky sexual behaviours ..are still rife’. Again important to note that this qualitative study cannot provide estimates of prevalence. Risky behaviours were reported by the participants which suggests that they could be prevalent, but this group of participants who attend a service may have higher prevalence of such behaviours. As noted above, it is important not to draw prevalence estimates from a small sample qualitative study.
Comments on the Quality of English LanguageNo major issues with english language. A few minor grammatical issues are identified in the report above.
Author Response

(The authors gave the same response as above.)

Round 2
Reviewer 2 Report
Comments and Suggestions for Authors
Comments
The authors have responded to all the comments made on the initial version, and have adopted the suggestions and made corrections as recommended.
However there one comment which has not been fully addressed.
Line 49 – 50 :It is unclear what the percentage figures refer to - % increase (compared to previous year ?) – not fully addressed - labelled as “STI rate” – what is the denominator ? eg as a percentage of swabs collected; or of patients attending with STI symptoms ??
Responses to the comments on the first version are noted below in yellow highlight italics.
Abstract
Does not mention the gender of the subjects- addressed
.Subjects stated as those aged 13 to 19 years – but gender distribution not mentioned - addressed
“Risky sexual behaviour .. is still rife”. Important to clarify that this qualitative study cannot estimate the prevalence of the ideas or practices – but rather explores the thinking behind the behaviours. Perhaps a more appropriate summary would be that belief in traditional or “natural” remedies is still strong, rather than scientific or western medical concepts.
No comment / response – but acceptable
Introduction
Line 48: Limpopo has also experienced . For readers not familiar with South Africa, it would be helpful to provide some further information on Limpopo – its geography, population etc - addressed
Line 49 : These are presumably districts within Limpopo – do they cover all the districts of Limpopo ? - addressed
Line 49 – 50 :It is unclear what the percentage figures refer to - % increase (compared to previous year ?) – not fully addressed - labelled as “STI rate” – what is the denominator ? eg as a percentage of swabs collected; or of patients attending with STI symptoms ??
Line 74: Is the clinic in province ? - addressed
More information provided on the NAYFS its functions and standards (line 70 – 96) which is helpful.
Additional statement of the objective for the study.
Materials and methods
Line 99-101: This comment repeats the comments in line 73-74 of the introduction – it is probably unnecessary to repeat it here, while retaining the comments in lines 73-74 - addressed
Line 103: It would be useful to include an estimate of the population in the clinic catchment, and/or the proportion or number of teenagers in the population. - addressed
Line 110: ‘productive education’ – probably should be ‘reproductive’ - addressed
Line 140-143: It is unclear what this sentence means. The first sentence states that teenagers who were not part of the Youth Friendly Service were excluded; and the second that 4 teenagers who ‘only’ attended the Youth Friendly Service were included. Did they only attend the Youth Friendly Service after interviews ? - addressed
Line 152: ‘trial-ran’ – suggest using the term ‘pre-tested’ - addressed
Line 189: ‘who experts’ - add ‘are’ – addressed (line 218)
Results
Table 1 revised to be clearer
Table 3 provides a useful summary of the key results- further clarified
4. Discussion
Line 327 Abstinence: while in this study sample, abstinence was rarely encountered, it is important to note that the sample was from teenagers attending the clinic and is not a community sample. It suggests that even teenagers at a young age have sexual contact, but qualitative studies cannot provide prevalence estimates, and a community sample is needed to determine the extent of this behaviour in the community. – addressed (line 371-75)
Similarly, line 339 also attempts to draw prevalence estimates from this qualitative study, which is not appropriate. – addressed with the addition of ‘some’ ( line 389)
The discussion of the effectiveness of yoghurt application to the vagina (lines 359-365) the consumption of ale vera juice (lines 368 – 381) or drinking of marula ( lines 383 ff) extends beyond the scope of the study on practices and behaviours. It would be more relevant to compare these beliefs and practices with the recommendations and advice that should be provided by the health or education facilities.
Addressed with comment that insufficient scientific evidence to support practice (line 400, 416, and 422)
4.2 Factors influencing choice of STI preventive measures
Lack of sex education (lines 414 ff). While the stated beliefs of the participants suggest a lack of sex education, there is no reporting of questions about sex education or access to sex education materials in the results section. It is perhaps an assumption.
Additional information provided
Accessibility of sexual health services. This is perhaps an issue that requires further research, as the study participants were accessed through a sexual health service, and thus do not represent the views of those who do not access such services. - Addressed
5. Conclusions
‘Risky sexual behaviours ..are still rife’. Again important to note that this qualitative study cannot provide estimates of prevalence. Risky behaviours were reported by the participants which suggests that they could be prevalent, but this group of participants who attend a service may have higher prevalence of such behaviours. As noted above, it is important not to draw prevalence estimates from a small sample qualitative study. -Addressed
Author Response
Dear Reviewer
Thank you for reviewing the manuscript for the second time. kindly note that the remaining point of clarity has been attended to as follows:
POINT BY POINT RESPONSES TO ROUND 2 REVIEWER Comments
The authors have responded to all the comments made on the initial version and have adopted the suggestions and made corrections as recommended.
However there one comment which has not been fully addressed.
Comment: Line 49 – 50: It is unclear what the percentage figures refer to - % increase (compared to previous year?) – not fully addressed - labelled as “STI rate” – what is the denominator? e.g. as a percentage of swabs collected; or of patients attending with STI symptoms??
Response: In line 50 -53 of the manuscript, authors has clarified the denominator of the prevalence rate, indicating that it was amongst patients attending with STIs symptoms.
All corrections made in the manuscript are highlighted in green.
Regards
Takalani
